# T-VecTTS: Adding time-varying-emotion control to flow-matching-based TTS

## Abstract

Recent advances in text-to-speech (TTS) have enabled natural speech synthesis, yet fine-grained, time-varying emotion control remains challenging. Existing methods typically provide only utterance-level control or require full-model fine-tuning with a large in-house emotional speech corpus. To overcome these shortcomings, we propose a *time-varying-emotion controllable* TTS (T-VecTTS), seamlessly adding the additional control to the pre-trained flow-matching-based TTS. To leverage the off-the-shelf model, we freeze the original model and attach a trainable branch that processes additional conditioning signals for emotion control. Moreover, we identify the flow step interval that is responsible for determining emotions and use it for detailed control. We further provide practical recipes for emotion control on three components: (1) an optimal layer choice via block-level analysis, (2) control scale during inference, and (3) selecting the temporal emotion window size. The advantages of our method include the zero-shot voice cloning capability, naturalness of the synthesized speech, and no need for a large emotional speech corpus or full-model fine-tuning. T-VecTTS achieves state-of-the-art emotion similarity scores (Emo-SIM and Aro–Val SIM).

## 1 Introduction

Recent advances in text-to-speech (TTS) technology have enabled highly natural and expressive speech synthesis, allowing users to generate lifelike utterances from arbitrary text (Zhao & Yang, 2023; Le et al., 2024; Kanda et al., 2024; Eskimez et al., 2024; Chen et al., 2024b; Du et al., 2024; Wang et al., 2017; Ren et al., 2019). As these systems mature, user demand for greater controllability, especially over emotional expression, has grown rapidly. While various approaches have been proposed to condition TTS on emotion, most existing methods (Zhao & Yang, 2023; Tang et al., 2024; Guo et al., 2023; Tang et al., 2023a; Zhou et al., 2022; Tang et al., 2023b; Lei et al., 2022; Shin et al., 2022; Lee et al., 2017; Li et al., 2021; Cai et al., 2021; Cho et al., 2024) support only utterance-level control and lack the ability to modulate time-varying emotional cues with fine temporal granularity within an utterance. This limitation is particularly critical in applications such as speech-to-speech translation, where accurate emotional transfer aligned over time is essential. In addition, most of them are trained on a limited number of speakers and samples (in an extreme case, only 1 speaker) due to the high cost of collecting a labeled emotion speech dataset, resulting in a lack of control over emotions with arbitrary speakers. This hinders their practical applicability.

Wu et al. (2024) shows remarkable success in time-varying emotion conditioning by fine-tuning pre-trained large-scale TTS. However, it 1) requires a huge training cost while fine-tuning the entire pre-trained model with 87k hours of a large-scale speech dataset, which includes 27k hours of an in-house emotion dataset, 2) compromises the performance of the pre-trained model, and 3) cannot intuitively adjust emotion intensity.

To address these limitations, we propose a time-varying-emotion controllable TTS (T-VecTTS) of any *zero-shot speaker*, seamlessly adding the additional control to the off-the-shelf model. Inspired by Zhang et al. (2023), we freeze the original large-scale pretrained model and attach a trainable copy of it, which controls time-varying emotion. It leverages priors learned by the off-the-shelf model for speech synthesis. In addition, we introduce an emotion-specific flow step interval for both training and inference. As shown in the Figure 1, emotion is established within a specific range of steps, so restricting control to this range improves expressiveness while reducing WER and computation.

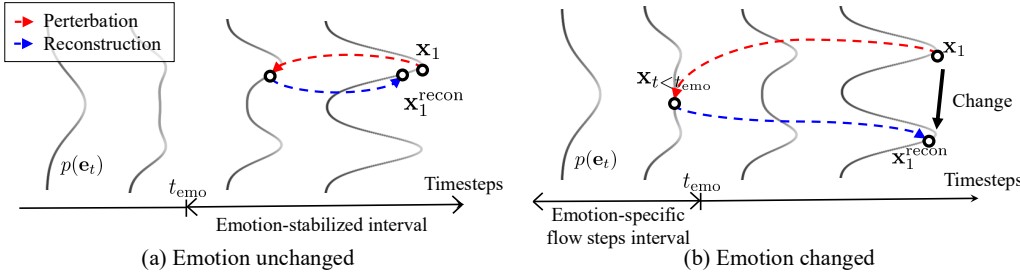

Figure 1: Emotion perturbation and reconstruction (a) without and (b) with emotion change on the distribution $p(\mathbf{e})$ across timesteps. (a) For flow steps $t > t_{\text{emo}}$, the model does not change the original emotion; for $t < t_{\text{emo}}$, the emotion changes. (b) The *emotion-specific flow steps interval* denotes the interval of steps where the emotion is allowed to change. Please see Section 4.1.1.

To investigate flow step–dependent emotion dynamics, we measure how the reconstructed emotion deviates from the original after perturbation. Lastly, we provide practical recipes for seamlessly integrating emotion control: 1) block-wise analysis of WER and speaker similarity, 2) control scale trades off between the behaviors of the base and the conditioned model, and 3) selection of an appropriate temporal window for emotion. Our approach preserves the ability of the original model and enables emotion control with a relatively small, publicly available emotional speech dataset ($\sim$400 hours) compared to the data used for the other comparable method ($\sim$87k hours) (Wu et al., 2024).

In conclusion, we demonstrate that desired control capabilities can be efficiently and effectively integrated into a large pre-trained system by freezing the original model and training only the conditioning branch, without requiring extensive computational resources or large, costly labeled datasets.

Our contributions are as follows.

- We propose to seamlessly add the additional control to large-scale & flow-matching TTS, enabling fine-grained, time-varying emotion control with zero-shot TTS.
- We identify an emotion-specific flow-step interval through an experiment on emotion dynamics across flow steps.
- We provide practical recipes to maximize a balance between word error and emotion control.
- Our approach preserves the quality, zero-shot capability, and naturalness of the original TTS model with careful design choices.

## 2 RELATED WORK

**Emotion-controllable text-to-speech** There have been a number of zero-shot TTS models (Chen et al., 2024a; Eskimez et al., 2024; Le et al., 2024) that enable voice cloning using a short reference audio clip. During the voice cloning stage, these models transfer the emotion expressed in the reference audio, allowing for natural emotion control. Cho et al. (2024) achieves emotion control by incorporating an additional emotion condition into the model. While previous studies have demonstrated the ability to control emotion at the utterance level, achieving fine-grained, time-varying emotion control remains a significant challenge. Wu et al. (2024) supports time-varying emotional control but requires full model finetuning, which incurs a substantial computational cost.

**Controlling large-scale generative model** Both flow-matching and diffusion models belong to closely related families of score-based generative models, which synthesize the target distribution by iteratively denoising random Gaussian noise. Accompanied by diffusion transformer (Rombach et al., 2021; Baevski et al., 2020; Peebles & Xie, 2023), they achieve huge success in scaling the generative model in both image (Rombach et al., 2021; 2022; Peebles & Xie, 2022; Labs, 2024; Esser et al.) and audio modalities, enabling product-level models. Based on the success, it has been introduced to add controllability to the pretrained model while leveraging the prior of the model with

fine-tuning (Zhang et al., 2023; Ye et al., 2023). In the image domain, Zhang et al. (2023) chooses a carefully designed conditional network by making a trainable copy only at downblocks of the original UNet and connecting it with the original branch through zero-convolution. It leads to huge success in spatial control. On the other hand, Avrahami et al. (2024) analyzes the influence of each layer in a DiT trained via flow matching, identifies the vital layers, and demonstrates the possibility of training-free editing during inference.

**Flow matching**   Flow matching (Lipman et al., 2023) learns a vector field $\mathbf{v}_t(\mathbf{x}; \theta)$ which transfer the known distribution (e.g., Gaussian noise) to the target distribution $p_1$, approximating the data distribution $q$ with a neural network $\theta$. In conditional flow matching (CFM), we have flow map $\psi_t(\mathbf{x}) = \sigma_t(\mathbf{x}_1)x + \mu_t(\mathbf{x}_1)$ conditioned on $\mathbf{x}_1$. With $\mu_0(\mathbf{x}_1) = 0$ and $\sigma_0(\mathbf{x}_1) = 1$, $\mu_1(\mathbf{x}_1) = \mathbf{x}_1$ and $\sigma_1(\mathbf{x}_1) = 0$, a vector field of the flow is $d\psi_t(\mathbf{x}_0)/dt = \mathbf{u}_t(\psi_t(\mathbf{x}_0) \mid \mathbf{x}_1)$. Then, CFM loss becomes :

$$\mathcal{L}_{\text{CFM}}(\theta) = \mathbb{E}_{t, q(\mathbf{x}_1), p(\mathbf{x}_0)} \left\| \mathbf{v}_t(\psi_t(\mathbf{x}_0)) - \frac{d}{dt}\psi_t(\mathbf{x}_0) \right\|^2. \tag{1}$$

Optimal transport further provides the OT-CFM loss:

$$\mathcal{L}_{\text{CFM}}(\theta) = \mathbb{E}_{t, q(\mathbf{x}_1), p(\mathbf{x}_0)} \left\| \mathbf{v}_t\left((1-t)\mathbf{x}_0 + t\,\mathbf{x}_1\right) - (\mathbf{x}_1 - \mathbf{x}_0) \right\|^2, \tag{2}$$

where the flow $\psi_t(\mathbf{x}_0)$ is interpolation between $\mathbf{x}_0$ and $\mathbf{x}_1$ with flow step $t \in [0, 1]$. With the learned $\mathbf{v}_t(\cdot; \theta)$, we synthesize $x_1$ from randomly sampled $\mathbf{x}_0 \sim N(0, I)$ by solving the ordinary differential equation (ODE) with the predicted flow $d\psi_t(\mathbf{x}_0)/dt$ from $t = 0$ to $t = 1$. As $t$ increases from 0 to 1 $\mathbf{x}_0$ is progressively transferred into $\mathbf{x}_1$; in practice this entails $K$ function evaluations (NFE), determined by the chosen solver and tolerance.

**Speech emotion recognition**   Speech Emotion Recognition (SER) has received increasing attention in recent years, with diverse modeling approaches proposed to better capture emotional cues from speech (Parthasarathy & Busso, 2017; Mirsamadi et al., 2017; Ghosal et al., 2019). One prominent line of work Wagner et al. (2023) leverages self-supervised learning to pretrain models on large-scale, unlabeled speech corpora. Models like wav2vec 2.0 (Baevski et al., 2020) are trained to learn contextualized audio representations without the need for emotion labels. These pretrained models are then adapted to downstream SER tasks by attaching a regression head or classifier and fine-tuning on emotion-labeled datasets. Since emotion labels are typically provided at the utterance level, the entire speech segment is passed through the pretrained encoder, and the resulting feature sequence is mean-pooled to obtain a single fixed-length representation. A regression head is trained on top of this representation to predict the corresponding emotion label for the utterance.

## 3   METHOD

In Figure 2, we present an overview of the training and inference processes of T-VecTTS. Given speaker reference audio $\mathbf{x}_{\text{spk}}$, emotion reference audio $\mathbf{x}_{\text{emo}}$, and target text $\mathbf{y}$, our goal is to synthesize speech that renders $\mathbf{y}$ in the *voice* of $\mathbf{x}_{\text{spk}}$ while following the *time-varying emotion trajectory* of $\mathbf{x}_{\text{emo}}$. For example, as shown in Figure 2 (b), when the emotion trajectory transitions from `sad` to `happy`, the synthesized speech exhibits a matching transition.

### 3.1   PIPELINE

Ours consists of two branches: 1) the original TTS branch and 2) the emotion conditioning branch.

**Original TTS branch**   Given a masked audio and a paired transcript, the original text-to-speech model (Chen et al., 2024a) learns an infilling task by predicting the masked part. Specifically, it requires a pair $(\mathbf{x}, y)$ where $\mathbf{x} \in \mathbb{R}^{F \times T}$ denotes an melspectrogram of an audio sample $\mathbf{s}$ and $\mathbf{y}$ is the corresponding transcript. $F$ is dimension of melspectrogram and $T$ is the sequence length. Given an input $\mathbf{x}$, we construct a noisy speech sample as $(1 - t)\mathbf{x}_0 + t\mathbf{x}_1$, and a masked speech sample as $(1 - \mathbf{m}) \odot \mathbf{x}_1$, where $t$ denotes the flow step, and $\mathbf{m} \in \{0, 1\}^{F \times T}$ is a random binary temporal mask. The noisy speech, masked speech, and transcript are concatenated before entering the model.

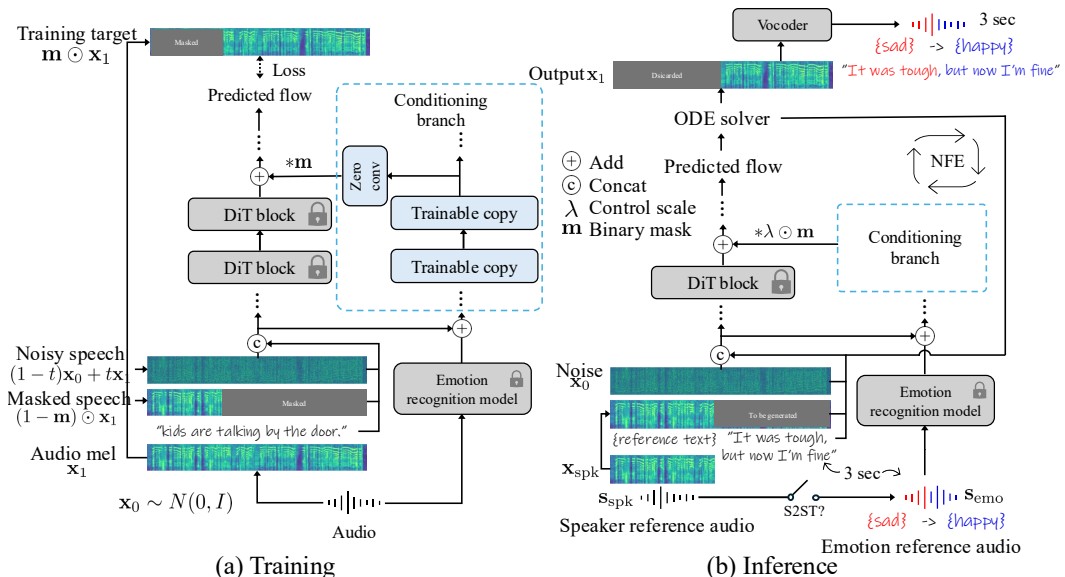

(a) Training  (b) Inference

Figure 2: **Overview of T-VecTTS.** (a) We train our model to infill the masked speech conditioned on the emotion via emotion conditioning branch (blue dashed box). (b) Our model generates speeches with varying emotion within an utterance, e.g., sad → happy. Please see Section 3.1

**Emotion conditioning branch**  As shown in the dashed box in Figure 2 (a), the conditioning branch builds upon a trainable copy of the pre-trained text-to-speech model. T-VecTTS connects the copied and the original model with zero-convolution (Zhang et al., 2023), which initially outputs zeros, thereby preserving the behavior of the original model while allowing the model to gradually incorporate additional conditioning. The conditioning branch consumes emotion $\mathbf{e} \in \mathbb{R}^{D_{\text{emo}} \times T}$, which is extracted using the speech emotion recognition model (SER) (Wagner et al., 2023), where each frame provides arousal and valence. We project the emotion embedding to match the channel dimension of the original DiT input embedding $F$ using a $1 \times 1$ convolution, and concatenate $[F; \text{proj}(\mathbf{e})]$ as the input to the conditioning branch.

The conditioning branch mirrors the base DiT stack, yielding a one-to-one correspondence between blocks. For $k$-th block, the conditioning output is passed through a zero-convolution and additively injected into the output of the corresponding base block. Let $\mathcal{F}_k(\cdot\,;\theta)$ and $\mathcal{F}_k(\cdot\,;\phi)$ be a $k$-th original DiT block and the corresponding conditioning block, respectively. Then, the final output $Z_{\text{new}}$ of the $k$-th block in the original network becomes

$$\mathbf{Z}_{\text{new}} = \mathcal{F}_k(\cdot\,;\theta) + \mathbf{m} \odot \mathcal{F}_k(\cdot\,;\phi). \tag{3}$$

For training, we prepare $(\mathbf{x}, \mathbf{y})$ training pairs with their the corresponding audio $\mathbf{a}$, and optimize only the conditioning block parameters $\phi$ using the following emotion-conditioned flow-matching loss, with conditioning $\mathbf{e}$:

$$\mathcal{L}_{\text{CFM}}(\phi) = \mathbb{E}_{t,q(\mathbf{x}_1),p(\mathbf{x}_0)} \left\| \mathbf{v}_t \left((1-t)\mathbf{x}_0 + t\mathbf{x}_1, \mathbf{e};\theta,\phi\right) - (\mathbf{x}_1 - \mathbf{x}_0) \right\|^2, \tag{4}$$

where $\mathbf{v}_t(\cdot\,;\theta,\phi)$ is now the combined model with the original model $\theta$ and conditional network $\phi$. The whole model is trained to predict the flow-derived target distribution $P\left(\mathbf{m} \odot \hat{\mathbf{x}} \mid (1-\mathbf{m}) \odot \hat{\mathbf{x}}, \mathbf{y}, \mathbf{e}\right)$.

### 3.2 EMOTION-SPECIFIC FLOW STEP

The original model is trained with the random flow step $t \sim [0, 1]$. Meanwhile, we opt to use only a subset of the interval while training with additional emotion embedding: $t \sim [0, t_{\text{emo}}]$, where $t_{\text{emo}} \in [0, 1]$ designates the range of employing our emotion branch. It achieves a lower word error rate (WER) and higher emotion similarity. We provide the supporting experiments in Section 4.1.1.

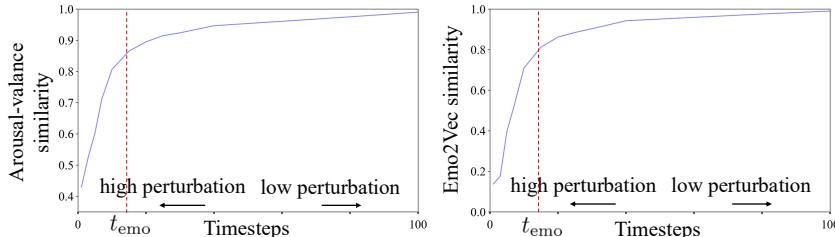

Figure 3: Reconstruction-after-perturbation supports this: with strong perturbation at $t < t_{\mathrm{emo}}$, the reconstructed audio scarcely preserves the original emotion. Please refer to Section 3.2.

### 3.3 SELECTIVE BLOCK

Rather than connecting all DiT blocks from the emotion branch to the main branch, we select only a subset based on their contribution to results. Specifically, to find the most critical blocks, we conduct a layer-wise analysis in Section 4.1.2. Each DiT block is bypassed via its residual connection, and we evaluate the resulting impact on both WER and speaker similarity. This analysis allows us to identify the most influential layers for stable control.

### 3.4 INFERENCE

As shown in Figure 2 (b), we prepare melspectrogram $\mathbf{x}_{\mathrm{spk}}$ from the speaker identity reference $\mathbf{s}_{\mathrm{spk}}$, the transcript $\mathbf{y}_{\mathrm{spk}}$ of the audio, target text $\mathbf{y}_{\mathrm{gen}}$, initial noise $\mathbf{x}_0 \sim N(0, I)$, and emotion $\mathbf{e} = \mathrm{SER}(\mathbf{s}_{\mathrm{emo}})$ from emotion reference audio $\mathbf{s}_{\mathrm{emo}}$. For Speech-to-Speech (S2ST) scenario, $\mathbf{s}_{\mathrm{spk}} = \mathbf{s}_{\mathrm{emo}}$. The original network takes three inputs $(\mathbf{x}_0, \mathbf{x}_{\mathrm{spk}}, \mathbf{y}_{\mathrm{text}})$ where $\mathbf{y}_{\mathrm{text}} = \mathbf{y}_{\mathrm{spk}} \| \mathbf{y}_{\mathrm{gen}}$ [1] and the conditioning branch takes fours inputs $(\mathbf{x}_0, \mathbf{x}_{\mathrm{spk}}, y_{\mathrm{text}}, \mathbf{e})$. At the output of each DiT block, we get $\mathbf{Z}_{\mathrm{new}}$ as follows:

$$\mathbf{Z}_{\mathrm{new}} = \mathcal{F}_k(\cdot\,;\,\theta) + \lambda \cdot \mathbf{m} \odot \mathcal{F}_k(\cdot\,;\,\phi), \tag{5}$$

where $\lambda$ represents the control scale. The control scale enables balancing between WER and emotion similarity. We provide the experiment according to various control scale in Section 4.1.3. The original network outputs a predicted flow field, which is used to solve an ODE. By integrating this flow starting from $\mathbf{x}_0$, we can synthesize speech $\hat{\mathbf{x}}_1$. Since the conditioning branch is trained only within the interval $[0, t_{\mathrm{emo}}]$, we apply it exclusively during this time range for inference. For the remaining steps, we set $\lambda = 0$.

## 4 EXPERIMENT

We provide an ablation study of our emotion control strategies in Section 4.1, training details in Section 4.2, and comparisons with prior work in Section 4.3.

### 4.1 ABLATION STUDY

For the ablation study, we used the RAVDESS as the evaluation data (Livingstone & Russo (2018)).

### 4.1.1 EMOTION-SPECIFIC FLOW STEP

We identify the emotion-specific flow step interval, which is the range of flow steps where the emotion of the generated speech is determined. As shown in Figure 1, in a flow-matching-based model, each flow step progressively transports a sample from a known prior distribution (Gaussian noise) toward the data distribution (e.g., natural speech). At early flow steps, the sample remains close to Gaussian noise, so even a minor transition can significantly affect the output. Conversely, as the flow step approaches 1, meaning that the sample is near the target distribution, perturbations have minimal impact on the final output.

---

[1] $\|$ denote sequence concatenation; $y_1 \| y_2 = \texttt{text1text2}$ as $y_1 = \texttt{text1}$ and $y_2 = \texttt{text2}$

Table 1: **Training flow step interval ablation study**

| timesteps | WER (%) ↓ | Emo-SIM ↑ | Aro-Val SIM ↑ |
|-----------|-----------|-----------|----------------|
| [0,1]     | 0         | 0.389     | 0.674          |
| [0,0.1]   | 1.9       | 0.565     | 0.876          |

(a) WER↓

(b) Speaker similarity↑

Figure 4: **Layer ablation with quantitative results.** Each dot marks (a) Word-Error-Rate and (b) Speaker similarity when a layer is skipped. Some layers greatly increase WER and decrease speaker similarity when skipped. Please see Section 4.1.2.

In this section, we present experiments designed to pinpoint the flow step interval where emotion is established. We prepare multiple *(audio sample, transcript)* pairs and, following the conditional flow matching training objective, interpolate between random Gaussian noise and audio at various flow steps to obtain intermediate flow maps. For each intermediate flow map, we resample and measure the emotional difference from the original audio. We measured using arousal-valence (Wagner et al., 2023) and Emo2Vec (Wagner et al., 2023).

As illustrated in Figure 3, when the flow step $t$ is close to 1, emotion barely changes. However, if we select a flow step $t$ near the Gaussian noise, the model fails to reconstruct the original audio, producing outputs with low emotion similarity. Through this process, we identify the emotion-specific flow step interval-where emotion similarity changes rapidly, $t \in \{0, t_{\text{emo}}\}$. Utilizing this interval during training and inference allows us to reduce word error rate (WER) and improve efficiency by excluding regions where emotion does not change from training. In Table 1, using the whole time step struggles to transfer emotion. See Appendix A for a more detailed explanation.

### 4.1.2 SELECTIVE BLOCK

A small subset of transformer blocks exerts a dominant influence on both textual fidelity and speaker identity preservation. Inspired by block-wise analyses in Stable Flow (Avrahami et al., 2024) for text-to-image, we propose a block-level analysis with transformer-based TTS models. As shown in Figure 4(a), we perform a block-level ablation on F5-TTS (22 residual blocks) by skipping one block at a time during inference and measuring the resulting WER. We repeat the experiment for speaker similarity in Figure 4(b). Skipping certain blocks substantially increases WER, and the same blocks also cause a notable drop in speaker similarity. This indicates that these blocks are critical to preserving both textual accuracy and speaker identity. Accordingly, we exclude layers 0, 1, 6, and 16 from connections with conditioning network and attach control only to the remaining blocks, which yields lower WER while preserving speaker and emotion similarity (Table Table 2).

### 4.1.3 CONTROL SCALE

During training, the control scale is fixed at 1.0, but during inference, it can be freely adjusted, allowing the model to interpolate between the behaviors of the original model and the condition-enhanced model. In this experiment, we investigate the impact of the control scale hyperparameter $\lambda$ on the TTS system. Specifically, setting $\lambda = 0$ activates only the base F5-TTS model, while $\lambda = 1.0$ fully integrates the output of conditioning network.

We measure WER, AutoPCP, SIM-O, Emo-SIM, and Arousal-Valence SIM. As shown in the table 3, we observe that increasing the control scale $\lambda$ leads to consistent improvements in the AutoPCP,

Table 2: **Selective blocks ablation study**

| Blocks | SIM-o ↑ | WER (%) ↓ | Emo-SIM ↑ | Aro-Val SIM ↑ |
|---|---|---|---|---|
| Full | 0.630 | 8.9 | 0.450 | 0.746 |
| Selective blocks | **0.684** | **0** | **0.565** | **0.872** |

Table 3: **Results with control scale**

| Control scale | SIM-o ↑ | WER (%) ↓ | Emo-SIM ↑ | Aro-Val SIM ↑ | AutoPCP ↑ |
|---|---|---|---|---|---|
| 0.0 | 0.579 | 2.9 | 0.692 | 0.845 | 3.62 |
| 0.1 | 0.589 | 0.63 | 0.724 | 0.864 | 3.68 |
| 0.3 | 0.590 | 6.56 | 0.735 | 0.887 | 3.69 |
| 0.5 | 0.595 | 7.92 | 0.734 | 0.892 | 3.69 |
| 1.0 | 0.593 | 9.58 | 0.742 | 0.892 | 3.67 |

Emo-SIM, and Arousal-Valence SIM metrics, showing that the control scale effectively enhances the emotional expressiveness of the generated audio. However, we also observe a substantial increase in WER as $\lambda$ increases. We attribute this to a trade-off between emotional expressiveness and linguistic clarity—stronger emotional modulation may introduce acoustic variations that degrade phoneme-level precision, thereby impacting word recognition accuracy.

### 4.1.4 EMOTION WINDOW SIZE

The speech emotion recognition model (Baevski et al., 2020) consists of Wav2vec and a subsequent regression head. A naive approach to obtain time-varying emotion is to feed each Wav2vec output token directly into the regression head, thereby producing an emotion label for each token. However, this method leads to a loss of emotional information and results in increased word error rate (WER). To address this, we adopt a window sliding interpolation strategy on the Wav2vec features. By applying this technique, we generate smoother and more context-aware features, which are then used as input to conditioning network. This approach preserves emotional information and decreases WER in Table 4. See Appendix B for a detailed explanation.

### 4.2 TRAINING

**Dataset**  Publicly available high-quality emotional TTS datasets are extremely scarce. In contrast to large-scale speech corpora, which typically range from 10K to 100K hours (He et al., 2024; Zen et al., 2019), existing emotional speech datasets are significantly smaller, often limited to just a few hundred hours (Lotfian & Busso, 2017; Richter et al., 2024; Busso et al., 2008). To overcome this limitation, we constructed a training dataset by combining six datasets (Busso et al., 2008; Richter et al., 2024; Zhou et al., 2021; Lotfian & Busso, 2017; Nguyen et al., 2023). Since we utilized all datasets with different characteristics, certain criteria were required to curate the training dataset. For the EARS dataset (Richter et al., 2024), we removed audio files exceeding 3 minutes because the baseline could not handle the audio length. Files consisting solely of non-verbal sounds, such as throat clearing, or lacking corresponding transcriptions, were excluded. Similarly, in the case of the Expresso dataset (Nguyen et al., 2023), multiple speakers' audio found in conversational datasets were omitted. Approximately 400 hours of data remained available for training. All files are upsampled or downsampled to 24kHz to meet the input file specifications of the baseline model.

Please see Appendix C for implementation details.

### 4.3 COMPARISONS WITH PRIOR WORKS

We compare T-VecTTS with flow-matching based zero-shot TTS models: VoiceBox (Le et al., 2024), ELaTE (Kanda et al., 2024), EmoCtrl-TTS (Wu et al., 2024), F5-TTS (Chen et al., 2024b). We also include SeamlessExpressive (Barrault et al., 2023). Since most of the model is not publicly available, we use the values reported by Emoctrl-TTS (Wu et al., 2024) except for F5-TTS. For experimental reliability, we faithfully follow the evaluation protocol to implement JVNV S2ST and EMO-Change.

Table 4: **Emotion window size ablation study**

| Emotion window size | WER (%) ↓ | Emo-SIM ↑ | Aro-Val SIM ↑ |
|---|---|---|---|
| 1 | 4.7 | 0.500 | 0.836 |
| 30 | **1.9** | **0.565** | **0.876** |

### 4.3.1 EVALUATION DATASET

Following Wu et al. (2024), we use two scenarios for evaluation: EMO-Change and JVNV S2ST. For EMO-Change, two different emotional audio clips, which are randomly sampled from the RAVDESS dataset, are concatenated. The RAVDESS dataset features English speech samples labeled with various emotions such as calm, happy, sad, angry, fearful, surprised, and disgusted. The concatenated audio enables the user to evaluate the explicit change of emotion over time. JVNV S2ST assumes Japanese-to-English speech-to-speech translation (S2ST). JVNV S2ST evaluates the emotion transferability of zero-shot TTS models in a cross-lingual setting. JVNV dataset contains emotionally expressive Japanese speech from four speakers (two male and two female) covering six emotions: anger, disgust, fear, happiness, sadness, and surprise. For S2ST translation, we first perform speech-to-text translation on the Japanese utterances to obtain English transcripts. These transcripts served as text prompts, while the original Japanese audio was used as the speaker and the emotion reference audio at the same time, as shown in Figure 2 (b). See Appendix D for more details.

### 4.3.2 EVALUATION METRIC

**Objective metric** Following the evaluation protocol of Wu et al. (2024), we utilize the same set of metrics to assess both the intelligibility and emotional controllability of the generated speech. We applied Whisper-Large (Radford et al., 2023) for automatic transcription and computing the corresponding ground truth texts. We express word error rate (WER) as a percentage. To assess prosodic similarity, we utilized AutoPCP (Barrault et al., 2023), a sentence-level metric that estimates the prosodic alignment between speech signals. We computed the AutoPCP score between the generated audio and the reference audio using AutoPCP multilingual v2.[2] We adopt Emo-SIM and Aro-Val SIM to evaluate whether the emotional content of the reference audio is well reflected in the synthesized speech. Emo-SIM measures similarity based on discrete emotional states using frame-level embeddings extracted by the Emotion2Vec model (Ma et al., 2024). Aro-Val SIM evaluates alignment in continuous emotion space—arousal and valence—estimated using a sliding window approach following Wagner et al. (2023). Together, these metrics provide a comprehensive assessment of emotional expressiveness in the synthesized speech.

**Subjective metric** We employed the following subjective evaluation metrics to assess the perceptual quality of the synthesized speech. SMOS (Speaker Similarity Mean Opinion Score) measures the perceived similarity between the speaker reference and the generated speech on a 5-point Likert scale, ranging from 1 (not at all similar) to 5 (extremely similar). NMOS (Naturalness Mean Opinion Score) evaluates the naturalness of the synthesized speech, where 1 indicates "bad" and 5 indicates "excellent." EMOS (Emotion Mean Opinion Score) assesses the emotional similarity between the emotion reference and the generated audio, rated from 1 (not at all similar) to 5 (extremely similar).

### 4.3.3 COMPARISON RESULTS

**Objective evaluation** In Table 5, we show the quantitative comparison with JVNV S2ST and EMO-Change datset. In both datasets, we achieve the best emotion similarity, including Emo-SIM and Aro-Val SIM, which supports that T-VecTTS effectively conveys the emotion in a reference audio. We also achieve competitive text fidelity performance in EMO-Change dataset. Since F5-TTS is only trained with english and chinese in Emilia101K (He et al., 2024), it is supposed to take input from the same languages shown during training, but not from japanese. The discrepancy may cause a lower WER compared to the reported value in the official repo of F5-TTS. Please note that we build T-VecTTS up on F5-TTS. We can find that T-VecTTS maintains the text fidelity of F5-TTS, the original model. We also preserve the speaker similarity of our base model in both datasets. Despite

---

[2]https://github.com/facebookresearch/seamless_communication

Table 5: **Quantitative comparison with JVNV S2ST and EMO-change dataset.**

| Model | Init | Training data (hours) | SIM-o ↑ | WER (%) ↓ | AutoPCP ↑ | Emo-SIM ↑ | Aro-Val SIM ↑ |
|---|---|---|---|---|---|---|---|
| | | JVNV S2ST | | | | | |
| (B1) SeamlessExpressive | - | - | 0.268 | **1.2** | 2.91 | 0.653 | 0.494 |
| (B2) VoiceBox (reproduction) | - | LL (60k) | 0.347 | 2.1 | 2.96 | 0.655 | 0.443 |
| (B3) ELaTE | B2 | LL (60k) + LAUGH (460) | 0.441 | 3.8 | 3.36 | 0.671 | 0.548 |
| (B4) VoiceBox (fine-tuned) | B2 | LL (60k) + IH-EMO (27k) + LAUGH (460) | 0.455 | 3.0 | 3.17 | 0.659 | 0.470 |
| (B5) EmoCtrl-TTS | B2 | LL (60k) + IH-EMO (27k) + LAUGH (460) | 0.448 | 4.4 | 3.38 | 0.693 | 0.647 |
| (B6) EmoCtrl-TTS(+) | B2 | LL (60k) + IH-EMO (27k) + LAUGH (460) | **0.497** | 3.2 | **3.50** | 0.697 | 0.643 |
| (B7) F5-TTS | - | Emilia (95k) | 0.459 | 4.5 | 2.87 | 0.684 | 0.627 |
| TTS-CtrlNet (Ours) | B7 | Public emotion speech (400) | 0.464 | 5.4 | 2.36 | **0.751** | **0.742** |
| | | EMO-change | | | | | |
| (B2) VoiceBox (reproduction) | - | LL (60k) | 0.600 | 1.2 | 3.31 | 0.685 | 0.663 |
| (B3) ELaTE | B2 | LL (60k) + LAUGH (460) | 0.643 | 0.2 | 3.52 | 0.700 | 0.761 |
| (B4) VoiceBox (fine-tuned) | B2 | LL (60k) + IH-EMO (27k) + LAUGH (460) | 0.622 | 1.1 | 3.31 | 0.678 | 0.655 |
| (B5) EmoCtrl-TTS | B2 | LL (60k) + IH-EMO (27k) + LAUGH (460) | 0.671 | **0.0** | 3.45 | 0.685 | 0.822 |
| (B6) EmoCtrl-TTS(+) | B2 | LL (60k) + IH-EMO (27k) + LAUGH (460) | **0.684** | 0.9 | 3.44 | 0.679 | 0.811 |
| (B7) F5-TTS | - | Emilia (95k) | 0.579 | 2.9 | 3.62 | 0.692 | 0.845 |
| TTS-CtrlNet (Ours) | B7 | Public emotion speech (400) | 0.589 | 0.6 | **3.68** | **0.724** | **0.864** |

Table 6: **MOS Comparison under JVNV S2ST and Emo-Change Conditions**

| Model | JVNV S2ST | | | Emo-Change | | |
|---|---|---|---|---|---|---|
| | SMOS | NMOS | EMOS | SMOS | NMOS | EMOS |
| F5-TTS | $2.0_{\pm 0.35}$ | $2.3_{\pm 0.44}$ | $2.6_{\pm 0.38}$ | $3.7_{\pm 0.25}$ | $3.8_{\pm 0.24}$ | $2.0_{\pm 0.35}$ |
| T-VecTTS | $\mathbf{2.7}_{\pm 0.33}$ | $\mathbf{3.2}_{\pm 0.37}$ | $\mathbf{3.4}_{\pm 0.40}$ | $3.6_{\pm 0.24}$ | $3.8_{\pm 0.22}$ | $\mathbf{3.3}_{\pm 0.25}$ |

being trained on a relatively small dataset (400 hours), ours achieves strong performance across multiple metrics, demonstrating the efficiency and effectiveness of our approach.

**Subjective evaluation**  We conducted a comparative evaluation of F5-TTS and T-VecTTS across SMOS, NMOS, and EMOS with JVNV S2ST and EMO-Change datasets. For each dataset, we collect 20 audio samples per model. Each MOS category is rated by a panel of 10 participants. The MOS results are well aligned with our quantitative results in Table 5. Specifically, both SMOS and NMOS indicate that T-VecTTS effectively preserves the speaker similarity and naturalness of the base model. Furthermore, the improvement in EMOS demonstrates that our approach enables seamless and robust emotion control. We provide additional user studies with other competitors in Appendix E. See Appendix F for the qualitative results.

In Appendix G, we report a matched-backbone comparison on F5-TTS by reproducing EmoCtrl-TTS under the same training recipe and dataset. In Appendix H, we analyze the effect of dataset size and type. The results suggest that the gains are attributable to our design choices rather than backbone differences or dataset parity.

## 5 CONCLUSION

In this paper, we propose a time-varying-emotion controllable TTS, seamlessly adding the control to the off-the-shelf model. Our approach enables the addition of desired conditions with relatively small public data and low training cost, without the need for full fine-tuning of the base model. We identify an emotion-specific interval within the multiple flow steps of a flow-matching-based model and leverage it during both training and inference to enhance performance. Furthermore, by analyzing the impact of each block within the DiT architecture, we selectively connect conditioning network and the original model, achieving superior results. We also propose an appropriate window size for extracting time-varying emotions from the emotion recognition model, which contributes to more natural and expressive speech synthesis. As a result, our model preserves the naturalness and zero-shot TTS capabilities of the original large-scale model, while significantly improving emotion control. Lastly, ours suggests the possibility of alleviating the burden of increasingly large AI systems by efficiently introducing fine-grained controllability, without extensive computation or large data.

**Limitations**  Despite the effectiveness of our approach, it is inherently limited by the capabilities of the underlying speech emotion recognition model (SER). Specifically, our SER cannot accurately recognize non-verbal cues (e.g., laughing, crying), which limits the transfer of non-verbal elements. For future work, we suggest exploring alternative conditions to support multiple and non-verbal conditions, thereby expanding the expressive range of controllable TTS systems.

**Ethics statements**   This work develops a TTS model with fine-grained, time-varying emotion control. We use only publicly available, properly licensed datasets; no human-subject experiments or personal/sensitive data are collected, and IRB approval was not required. We acknowledge risks of misuse (e.g., voice spoofing or deceptive media) and outline mitigations such as consent-based usage, provenance/watermarking, and usage restrictions; any released artifacts will comply with licenses and exclude PII. We assess basic fairness considerations and confirm compliance with relevant privacy and IP laws, and we adhere to the ICLR Code of Ethics.

**Reproducibility Statement**   We provide full implementation details of our proposed TTS model, including model architecture, training hyperparameters, and evaluation protocols in the main paper and appendix. Data sources and preprocessing steps are clearly described, and we only use publicly available datasets. Additional ablation studies and analysis are included to support robustness. To facilitate reproducibility, we supply anonymized source code and scripts for training and inference as supplementary material.

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

## A MORE EXPLANATION OF EMOTION-SPECIFIC FLOW STEP

Section 4.1.1 finds the flow steps controlled to reflect the reference emotion. We name these steps "the emotion-specific flow step". Specifically, we investigate *reconstruction fidelity* where a real audio sample is given. First, we apply interpolation with random noise to the original audio $x_0$ to compute intermediate latent flows at various flow steps. We then evaluate how well the emotional content can be reconstructed from these flows using two metrics: Arousal-Valence similarity (Aro-Val sim) and Emo2Vec similarity. We define flow steps that result in low emotion similarity as emotion-specific flow steps, since they show poor emotion reconstruction. Our key finding is that emotion control is most effective when applied at these step.

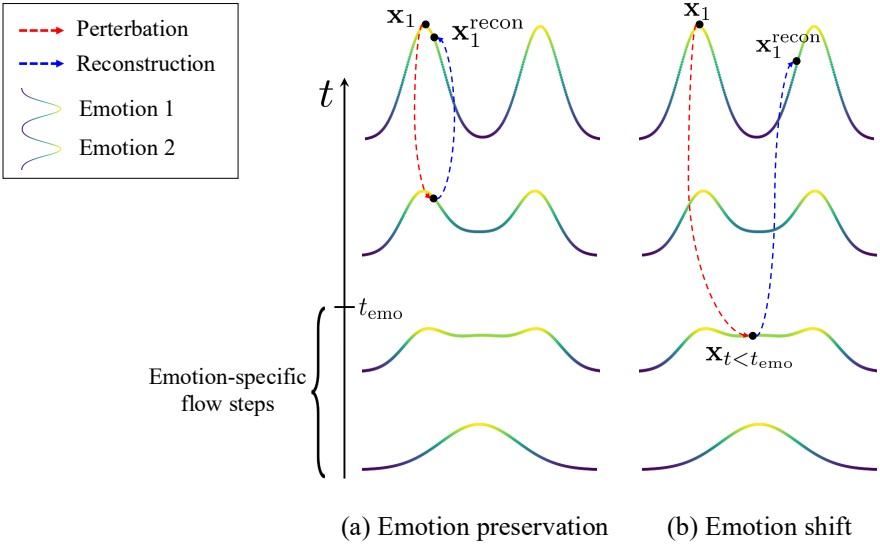

(a) Emotion preservation   (b) Emotion shift

Figure 5: **Illustration of emotion-specific flow steps.** (a) Slight perturbation of $x_0$ preserves emotion similarity. (b) Perturbation beyond the emotion-specific flow step leads to failed emotion reconstruction and emotion shift.

Figure 5 (a) shows a scenario where emotion reconstruction succeeds. When perturbing $x_0$ along a flow path closer to the target emotion distribution (as opposed to pure noise), the reconstructed audio $x$recon remains emotionally aligned with $x_0$, resulting in high similarity. Figure 5 (b) shows a failure case. Perturbing $x_0$ in a noise-like direction moves the intermediate representation away from the correct emotion, leading $x_{\text{recon}}$ to align with a different emotional distribution.

**Evaluation samples**   To evaluate this, we sample 120 speech audio clips from the RAVDESS dataset, ensuring a diverse range of emotional expressions. We use F5-TTS as the TTS model and, for each audio sample, performed interpolation with six different random noise vectors to generate intermediate flows.

**Computational cost**    The use of emotion-specific flow steps reduces computational overhead during inference, thereby improving the overall generation speed. On an NVIDIA RTX A5000, we measured the inference time required to generate a single audio sample of fixed length, using the same reference audio and text input across settings. As shown in Table 7, applying CtrlNet only to the early flow steps $[0, \ 0.1]$ introduces a small overhead compared to the baseline, while applying it to the full range $[0, \ 1]$ increases inference time more noticeably. This demonstrates that partial control offers a good trade-off between performance and efficiency.

These values were averaged over 20 runs to ensure consistency. The results demonstrate that partial application of CtrlNet in early flow steps is more efficient than full-step control, effectively balancing inference speed and emotional expressiveness.

Table 7: Inference time (seconds) with different CtrlNet application ranges.

| CtrlNet application range | Inference time (s) |
|---|---|
| No CtrlNet (baseline) | 3.708 |
| CtrlNet on $[0, \ 0.1]$ | 4.212 |
| CtrlNet on $[0, \ 1]$ | 5.416 |

## B    MORE EXPLANATION OF EMOTION WINDOW SIZE

In Section 4.1.4, we propose not to use emotion window size = 1 in the speech emotion recognition (SER) model Wagner et al. (2023). For better clarity, we also include a corresponding Figure 6 to illustrate this setup. During training, the SER model is trained on full utterances paired with a single emotion label, since it is challenging to annotate time-varying emotions. For a given audio input, the model uses wav2vec to produce a sequence of token-level features, averages them, and then passes the result through a regression head to predict a single emotion. At inference time, we mirror this

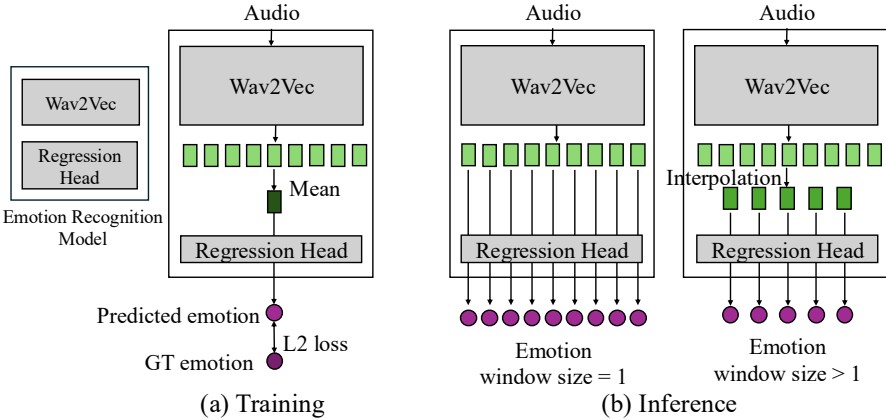

(a) Training  (b) Inference

Figure 6: **Emotion window step** Using window size = 1 fails to convey emotion properly, while setting a proper window size improves prediction consistency

training setup by applying interpolation over wav2vec outputs using an emotion window size greater than 1, effectively approximating the averaging process. This leads to improved word error rate (WER) and emotion similarity compared to using no interpolation (i.e., window size = 1), as shown in Table 2.

## C    IMPLEMENTATION DETAILS

We use F5-TTS Chen et al. (2024a) as our baseline model. Although F5-TTS is trained only on English, it supports cross-lingual zero-shot voice cloning and can use speech from other languages as reference audio. However, it struggles with time-varying emotional control and often produces

Table 8: Details of training dataset

| Dataset | Duration (hours) | # of speakers | # of emotions | Modality |
|---|---|---|---|---|
| IEMOCAP | 12.44 | 10 | 8 | Audio-Visual |
| ESD | 29.07 | 10 | 5 | Audio |
| MSP-Podcast | 319.15 | ∼500 | 7 | Audio |
| Expresso | 11.26 | 8 | 4 | Audio |
| RAVDESS | 1.48 | 24 | 8 | Audio-Visual |
| EARS | 54.96 | 20 | 6 | AUdio |
| Total | 428.36 | | | |

utterances with a monotonic emotional trajectory, which leads to unnatural or robotic-sounding speech. T-VecTTS preserves the cross-lingual zero-shot capability by freezing the original network, while additionally providing time-varying emotional control. We believe that introducing natural emotion transferability helps resolve the monotonicity issue. We use a wav2vec-based emotion recognition model (Wagner et al., 2023), which predicts arousal-valence-dominance. Following Wu et al. (2024), we exclude dominance and use only arousal-valence. Emotion recognition model consists of two subsequent network, wav2vec (Baevski et al., 2020) and a regression head. First, we get the token length corresponding to the length of the given audio. The token is interpolated with the emotion window size $W_{\text{emo}}$ and returns $e \in \mathbb{R}^{D_{\text{emo}} \times T}$. After that, $\mathbf{e}$ goes through $1 \times 1$ convolution layer, resulting in $\mathbf{e} \in \mathbb{R}^{F \times T}$. We use set 1e-5 as a learning rate and train with two A5000 GPUs with 8000 batch frames (the length of melspectrogram) up to 24000 steps for each configuration. We provide details of the training dataset in Table 8.

## D  EVALUATION DATASET

**EMO-Change**  We follows Wu et al. (2024) to implement the EMO-Change dataset using the RAVDESS dataset Livingstone & Russo (2018) to evaluate the model's ability to synthesize speech with nuanced emotional transitions, which features English speech samples labeled with emotions such as calm, happy, sad, angry, fearful, surprised, and disgusted. RAVDESS provides two transcripts - *"kids are talking by the door"* and *"dogs are sitting by the door"*- each spoken with various emotional intensities. For the EMO-Change dataset, we randomly choose two *"kids are talking by the door"* utterances, each expressing a different emotion, remove any silent segments, and concatenate them to form a single audio prompt. During evaluation, this concatenated audio is used as the emotional reference, while the repeated sentence *"dogs are sitting by the door dogs are sitting by the door"* serves as the text prompt for the zero-shot TTS model. This design tests the model's capability to reproduce the emotional shifts from the audio prompt in its generated speech, while maintaining the target text and speaker identity.

**JVNV S2ST**  For speech-to-speech translation, we follow the Japanese-to-English speech-to-speech translation (S2ST) experimental protocol of Wu et al. (2024) to evaluate the emotion transferability of zero-shot TTS models in a cross-lingual setting. Specifically, we use the JVNV dataset, which contains emotionally expressive Japanese speech from four speakers (two male and two female) covering six emotions: anger, disgust, fear, happiness, sadness, and surprise, as well as diverse nonverbal vocalizations. For S2ST translation, we first perform speech-to-text translation on the Japanese utterances to obtain English transcripts. These transcripts served as text prompts, while the original Japanese audio was used as the audio prompt for the zero-shot TTS model, from which we extracted both nonverbal and emotion embeddings. Since our base model Chen et al. (2024a) is not trained on Japanese and English within a model, we transform the Japanese transcript into its phonetic representation using Romaji[3]. Given a transcript and the corresponding source audio, the model generates English speech that preserves the speaker's identity and emotional characteristics from the source. We evaluate the similarity of speaker and emotion between the original Japanese speech and the synthesized English output using objective and subjective metrics, as described in the following section.

---

[3]https://en.wikipedia.org/wiki/Romanization_of_Japanese

# E    ADDITIONAL USER STUDY

We provide the actual survey page used in our study in Figure 7.

In the main paper, we only included a comparison with F5-TTS and excluded other competitors due to the limited availability of their models and speech samples. Since most competitors are not publicly accessible, we could not collect a sufficient number of audio samples for a comprehensive comparison. To address this, we present a qualitative evaluation in this section based on a limited set of samples, using three perceptual metrics: SMOS (Speaker similarity Mean Opinion Score), NMOS (Naturalness Mean Opinion Score), and EMOS (Emotion Mean Opinion Score).

The evaluation is conducted in two independent parts, corresponding to the two datasets described earlier. Part 1 compares 11 Japanese reference audio samples with English audio generated by our model and five baseline models. Part 2 compares English reference samples (8 utterances) against outputs from our model and four baselines. All ratings were collected on a 5-point Likert scale (1–5), and consistent with the main paper, each MOS category was evaluated by 10 participants.

**Subjective evaluation**    We provide the quantitative results within 6 models in Table 9. The values in parentheses next to each score represent the 95% confidence intervals. In terms of EMOS, ours ranks 1st on JVNV-S2ST and 2nd on EMO-Change, showing comparable performance to EmoCtrl-TTS. Considering the confidence intervals, the difference between 1st and 2nd place is not statistically significant. Compared to its baseline model F5-TTS, our method achieves substantial improvements in emotional similarity, with EMOS increasing from 2.91 to 3.55 on JVNV-S2ST and from 2.69 to 4.02 on EMO-Change. Furthermore, ours maintains or improves upon F5-TTS in both SMOS and NMOS, and also outperforms or matches other baselines within the margin of error, indicating strong overall performance across all metrics.

Table 9: **More quantitative comparison with JVNV S2ST and EMO-change dataset.**

| ID | JVNV S2ST | | | | EMO-Change | | |
|---|---|---|---|---|---|---|---|
| | Model | SMOS↑ | NMOS↑ | EMOS ↑ | Model | SMOS ↑ | NMOS ↑ | EMOS ↑ |
| B1 | SeamlessExpressive | $2.19_{\pm0.40}$ | $3.18_{\pm0.40}$ | $3.05_{\pm0.37}$ | | | | |
| B2 | VoiceBox | $2.81_{\pm0.36}$ | $3.06_{\pm0.46}$ | $3.35_{\pm0.35}$ | VoiceBox | $4.05_{\pm0.25}$ | $3.96_{\pm0.27}$ | $2.57_{\pm0.43}$ |
| B3 | Elate | $3.27_{\pm0.30}$ | $2.89_{\pm0.44}$ | $3.51_{\pm0.32}$ | Elate | $4.26_{\pm0.28}$ | $4.26_{\pm0.23}$ | $3.52_{\pm0.27}$ |
| B4 | EmoCtrl-TTS | $2.87_{\pm0.34}$ | $3.03_{\pm0.48}$ | $3.26_{\pm0.39}$ | EmoCtrl-TTS | $4.24_{\pm0.24}$ | $4.14_{\pm0.22}$ | $4.18_{\pm0.21}$ |
| B5 | F5-TTS | $2.26_{\pm0.38}$ | $2.91_{\pm0.47}$ | $2.91_{\pm0.40}$ | F5-TTS | $3.93_{\pm0.25}$ | $3.93_{\pm0.25}$ | $2.69_{\pm0.40}$ |
| – | T-VecTTS(Ours) | $2.90_{\pm0.36}$ | $3.68_{\pm0.34}$ | $3.55_{\pm0.40}$ | T-VecTTS(Ours) | $3.96_{\pm0.23}$ | $4.10_{\pm0.25}$ | $4.02_{\pm0.29}$ |

# F    QUALITATIVE RESULTS

We provide qualitative results in *"supple_demo/index.html"*, which can be accessed after extracting the attached *"supplementary.zip"* file. Since ELaTe Chen et al. (2024b), Voicebox Le et al. (2024), and EmoCtrl-TTS Wu et al. (2024) are not publicly available, all the qualitative results of competitors are from the official project page of EmoCtrl-TTS, excluding F5-TTS.

With EMO-Change, the reference audio is a concatenation of two utterances with different emotions, so we expect the generated audio to reflect both emotions in sequence. However, Seamless, ELaTE, Voicebox, and F5-TTS tend to generate speech with a single dominant emotion rather than a sequential transition. In contrast, our method captures both emotions well, aligning them sequentially with natural prosody and speaker similarity comparable to or better than the backbone. EmoCtrl-TTS also performs similarly well. We note that F5-TTS suffers from artifacts while synthesizing high-pitch speeches (e.g., an overly excited female voice). Since T-VecTTSuses F5-Tts as a backbone, both F5-TTS and ours suffer from the results of "Happy-¿Disgusted" (a) with the EMO-Change dataset.

With JVNV-S2ST, our method effectively reflects time-varying emotional changes. For example, in "angry" (b), the emotion intensifies toward the end of the utterance. The generated speech remains natural, with no degradation in speaker similarity, and often even yields better results. Some outputs show atypical English pronunciation, which is expected since the baseline model, F5-TTS, is trained only on Chinese and English, not Japanese. As a result, when the input audio is in Japanese, traces of Japanese prosody can appear. Our model occasionally shows similar tendencies.

Notably, EmoCtrl-TTS utilizes both the emotion encoder and NV encoder, while ours only uses the emotion encoder. Therefore, please do not account for the transfer of non-verbal cues such as laughter or sobbing.

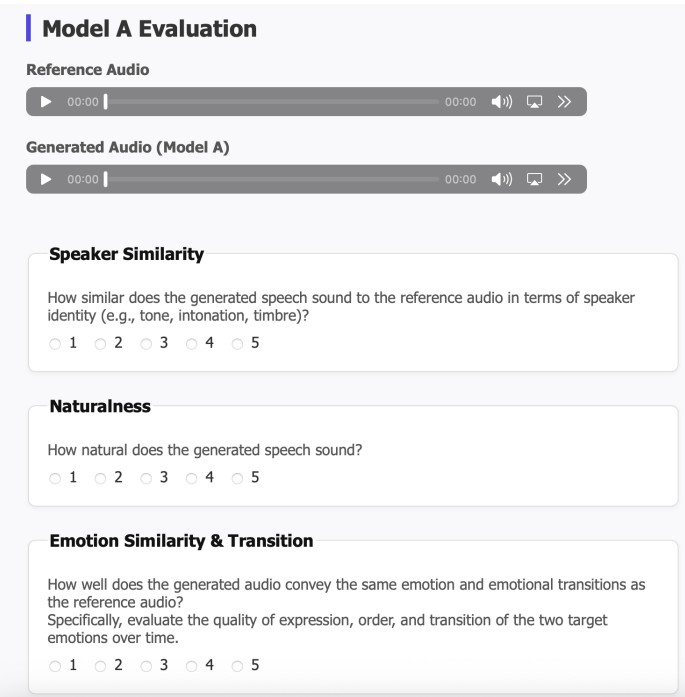

Figure 7: **Survey page**

## G  COMPARISONS WITHIN THE SAME BASELINE MODEL & DATASET

To address the fairness concern, we finetuned a pretrained [F5-TTS] using the same 400-hour emotion dataset. The resulting model showed minimal improvement in emotion similarity and speaker similarity, and even a slight degradation in WER, as shown in the Table 10. This suggests that merely exposing [F5-TTS] to the same data is not sufficient to improve controllability or fine-grained emotional expressivity.

Table 10: Comparisons within the same baseline model & dataset

| Methods | Backbone | Training data (hours) | SIM-o ↑ | WER (%) ↓ | AutoPCP ↑ | Emo-SIM ↑ | Aro-Val SIM ↑ |
|---------|----------|----------------------|---------|-----------|-----------|-----------|---------------|
| JVNV S2ST | | | | | | | |
| F5-TTS | N/A | Emilia (95k) | 0.459 | 4.5 | 2.87 | 0.684 | 0.627 |
| EmoCtrl-TTS (reproduce) | F5-TTS | Public emotion speech (400h) | 0.443 | 7.8 | 2.31 | 0.692 | 0.659 |
| TTS-CtrlNet (Ours) | F5-TTS | Public emotion speech (400h) | 0.464 | 5.4 | 2.36 | 0.751 | 0.742 |
| EMO-Change | | | | | | | |
| F5-TTS | N/A | Emilia (95k) | 0.579 | 2.9 | 3.62 | 0.692 | 0.845 |
| EmoCtrl-TTS (reproduce) | F5-TTS | Public emotion speech (400h) | 0.572 | 4.4 | 3.69 | 0.709 | 0.851 |
| TTS-CtrlNet (Ours) | F5-TTS | Public emotion speech (400h) | 0.589 | 0.6 | 3.68 | 0.724 | 0.864 |

## H  DATASET SIZE/TYPE ABLATIONS

To better understand the relationship between dataset size/type and performance, we conducted additional ablation studies under the same compute budget:

- Emotion speech data (400h → 200h)

- Non-emotional data (Panayotov et al. (2015) 400h)

As shown in the Table 11, reducing the emotion dataset from 400h to 200h leads to a small increase in WER and slightly lower emotion similarity, but the drop is minor. In contrast, when using non-emotional LibriSpeech data—a read-speech dataset with minimal emotional variation—we observed a degradation in emotional metrics, particularly Emo-SIM and Aro-Val SIM. This highlights the importance of emotion-rich supervision for our task. Furthermore, all variants were trained with the same number of total steps to ensure a fair comparison in terms of compute cost.

Table 11: Impact of dataset size and composition on model performance

| Methods | Backbone | Training data (hours) | SIM-o ↑ | WER (%) ↓ | AutoPCP ↑ | Emo-SIM ↑ | Aro-Val SIM ↑ |
|---------|----------|----------------------|---------|-----------|-----------|-----------|---------------|
| JVNV S2ST | | | | | | | |
| F5-TTS | N/A | Emilia (95k) | 0.459 | 4.5 | 2.87 | 0.684 | 0.627 |
| | | LibriSpeech (400h) | 0.461 | 3.1 | 2.44 | 0.692 | 0.635 |
| T-vecTTS (Ours) | F5-TTS | Public emotion speech (200h) | 0.464 | 5.9 | 2.39 | 0.744 | 0.734 |
| | | Public emotion speech (400h) | 0.464 | 5.4 | 2.36 | 0.751 | 0.742 |
| EMO-Change | | | | | | | |
| F5-TTS | N/A | Emilia (95k) | 0.579 | 2.9 | 3.62 | 0.692 | 0.845 |
| | | LibriSpeech (400h) | 0.581 | 1.2 | 3.64 | 0.694 | 0.854 |
| T-vecTTS (Ours) | F5-TTS | Public emotion speech (200h) | 0.575 | 1.9 | 3.55 | 0.717 | 0.865 |
| | | Public emotion speech (400h) | 0.589 | 0.6 | 3.68 | 0.724 | 0.864 |

