# OpenReview forum: "T-VecTTS: Adding time-varying-emotion control to flow-matching-based TTS"
_ICLR.cc/2026/Conference — Submitted to ICLR 2026_

### Official Review · Reviewer_fYW3 · 2025-10-21

**Soundness:** 1
**Presentation:** 2
**Contribution:** 2
**Rating:** 4
**Confidence:** 5

**Summary:**

This paper proproses a method to add fine-grained, time-varying emotion control to a pre-trained flow-matching TTS system by 1) attaching a trainable conditioning branch connected via zero-convolutions for gradually incorporate additional conditioning, and by 2) restricting control to an identified “emotion-specific flow-step” interval.

**Strengths:**

- This paper demonstrates that emotion is determined in the early flow steps, which is intuitive. The proposed emotion-specific flow-step concept (restricting conditioning to [0, tₑₘₒ]) is well-motivated and may have broader implications for controlling other attributes in flow or diffusion models. The reconstruction-after-perturbation experiments effectively support this observation.

- This paper presents a computationally efficient approach for introducing fine-grained, time-varying emotional control into large pre-trained TTS systems.

- This paper includes multiple ablations (flow step range, selective blocks, control scale, emotion window size) that evaluate design choices and tradeoffs (emotion vs. WER, inference time).

**Weaknesses:**

- The primary weakness of this work lies in the statistical validity of its experimental setup. As detailed in Appendix D, the test set appears to contain an extremely small number of samples, which raises significant concerns about the generalizability of the conclusions. This limited scale is the likely explanation for seemingly implausible results, such as the 0% WER reported in Tables 1, 2, and 5. A perfect WER is highly improbable in any realistic TTS evaluation, given the inherent recognition noise of even strong ASR models like Whisper Large-v3. **Consequently, findings based on such a small-scale testbed lack statistical significance and may not reflect the model's true performance.**

- The MOS scores in Table 6 are reported but lack confidence intervals and statistical tests.

**Questions:**

- Please clarify the WER = 0 entries (Tables 1/2/5). This is the most critical issue, as it makes the experimental results hard to trust.

- Please add confidence intervals and statistical tests for the MOS scores reported in Table 6.

- Please fix some presentation issues:
  1) in line 423, typo "datset"
  2) in Table 6, there is a missing \toprule line.
  2) in Table 10, there is a missing \bottomrule line.

- ICLR appendix usually does not include a title, considering removing it.

---

> ### Author Response · Authors · 2025-11-25
>
> ## W1. Number of samples
> Thank you for pointing this out. We clarify the origins and validity of the WER = 0 entries.
>
> (1) The WER=0 in Table 5 is not produced by us.
> It is directly quoted from the original EmoCtrl-TTS paper. Table 5 reports competitor results exactly as provided in their publication.
>
> (2) Tables 1 and 2 were evaluated on the RAVDESS dataset using 192 test samples.
> Specifically, our evaluation consists of:
> 	•	8 emotions
> 	•	8 speakers (4 male, 4 female)
> 	•	3 random seeds
> 	•	1 fixed prompt (“dogs are sitting on the floor”)
>
> We will explicitly include this information in Appendix D.
>
> (3) Text normalization was applied prior to WER computation.
> This includes standard processing steps such as punctuation removal, casing normalization, and whitespace handling, which prevent spurious mismatches in the ASR output.
>
> (4) Reference and generated speech are in the same language.
> Since our evaluation is not cross-lingual, no out-of-distribution effects occur that typically increase ASR errors. This contributes to stable and low WER values.
>
> We hope these clarifications address the concern regarding test-set size and the plausibility of the WER results. We will revise Appendix D accordingly and acknowledge the value of additional prompts and seeds for future robustness analysis.
>
> ## Q2. Confidence interval
> Thank you for the constructive feedback. We have now added confidence intervals for all MOS evaluations, which we believe strengthen the statistical validity of our results. The values are as follows:
>
> ### JVNV S2ST
> | Model    | SMOS | NMOS | EMOS |
> | -------- | -------- | -------- | -------- |
> | F5-TTS| 2.0±0.35 | 2.3±0.44     | 2.6±0.38     |
> | T-VecTTS| 2.7±0.33| 3.2±0.37  | 3.4±0.40|
>
> ### EMO-Change
> | Model    | SMOS | NMOS | EMOS |
> | -------- | -------- | -------- | -------- |
> | F5-TTS| 3.7±0.25 |  3.8±0.24   | 2.0±0.35|
> | T-VecTTS| 3.6±0.24| 3.8±0.22 | 3.3±0.25|
>
> These confidence intervals have been included in the revised paper.
>
>
> ## Q3. Presentation issue
> Thank you for pointing this out. We have corrected the presentation issues in the revised version of the paper.

---

> > ### Comment · Reviewer_fYW3 · 2025-11-25
> >
> > Thank the author for the additional clarification, which helps me understand that the WER=0 result is reasonable due to the small number of test samples (192 in total), the very clean audio, and the identical text content.
> >
> > On one hand, I’ve decided to raise the Overall Rating to 6 and set Soundness to 2, acknowledging that the reported results are plausible while still keeping concern about the model’s generalization ability given such a small test set. On the other hand, I have lowered my Confidence score to 3, since after reviewing comments from others who are more familiar with these test sets, I feel my original review is comparatively optimistic.
> >
> > I don’t have further questions at the moment. I hope my suggestions will be helpful for improving the quality of the paper.

---

> > > ### Author Response · Authors · 2025-11-26
> > >
> > > Thank you for the follow-up and for reconsidering the ratings.
> > > We’re glad the clarification was helpful, and we acknowledge the concern regarding the small test set. In the revision, we will more clearly describe the evaluation setup, discuss its limitations, and incorporate suggestions raised by you and the other reviewers to strengthen the analysis.
> > >
> > > Your feedback has been very helpful for improving the paper. Thank you again for your constructive comments.

---

### Official Review · Reviewer_VQtb · 2025-10-30

**Soundness:** 3
**Presentation:** 3
**Contribution:** 3
**Rating:** 4
**Confidence:** 4

**Summary:**

T-Vec TTS proposed to add time varying emotion control to pretained flow matching based TTS without degrading the base model's performance. Instead of retraining or fine-tuning a whole mode on emotional data, T-Vec TTS rather freezes a large pretrained zero shot TTS  and attaches a lightweight conditional branch to handel emotion input. This design brings several key novelties. In essence, T-VecTTS's technical innovation lies in how the emotion is integrated: by isolating when (which diffusion step) and where (which model layers) to apply the control, and doing so via a non-disuptive, additive branch. This contrasts with earlier approaches that either fine-tuned entire models (risking overfitting and quality loss) or applied controls in a coarse way

**Strengths:**

(1) T-VecTTS inherits zero-shot  capability from its base model. It can use a short audio prompt of any new speaker to clone that voice. Additionally, it adds emotion control on top of zero-shot TTS.

(2) No full fine tuning by freezing the original model and training only a small branch, T-VecTTS avoids the heavy cost of full model fine-tuning.

(3) Due to zero-initialized branch and selective conditioning, T-VecTTS manages to retain the high naturalness and intelligibility of the original TTS.

(4) T-VecTTS achieved state-of-the-art scores on emotion similarity metrics (Emo-SIM and Aro-Val SIM) compared to prior systems.

(5) In brief the strenghts of  T-VecTTS is to provides a practical and efficient solution: adding powerful emotion controls to a large pre-trained TTS without needing massive data or sacrificing the original model’s quality.

**Weaknesses:**

Following are some of my concerns

(1) Generalization : (a) The generalization is tied to the pre-trained base model's capabilities. While it preserves the original model's naturalness and speaker identity in most cases, it can inherit the base model's weaknesses.  For example, the F5-TTS backbone struggles with extreme vocal expressions  both F5 and T-VecTTS produce artifacts for an overly excited high-pitched voice.

(b) When transferring emotion from Japanese audio into English speech, the output sometimes carries residual Japanese prosody (an accent) because the base was trained only on English/Chinese. These cases show that T-VecTTS may falter with out-of-distribution inputs (unusual pitch ranges, unseen language prosody), indicating incomplete robustness despite its zero-shot design.

(c) Since it use F5-TTS as backbone my concern is  adapting T-VecTTS to a different TTS architecture would require a similar analysis to find where to inject control without breaking fidelity. If a TTS model doesn’t have a clear emotion-specific stage, applying this approach could be non-trivial.

(2) Dependence on SER:  Although authors have higlighed this in limitation section, a core limitation of T-VecTTS lies in its dependence on an external speech emotion recognition model for quided emotions. The model only controls the emotional trajectory that the SER can detect  in case of T-VecTTS, frame-wise arousal and valence values. This means emotional expression is encoded in just two dimensions, which restricts the diversity of emotions that can be distinctly conveyed. Subtle or complex emotions that share similar arousal/valence (e.g. fear vs. anger, which are both high arousal and negative valence) might not be well distinguished, since no discrete emotion category label is explicitly used. Moreover, the reliance on SER makes T-VecTTS inherently limited by the recognizer’s accuracy and scope.

(3) Experiments:

(a) Not all baselines were reproduced some were taken as reported in prior papers, whereas T-VecTTS was evaluated on presumably newly generated audio, which might not be perfectly comparable.

(b) There were no preference tests or qualitative feedback on the emotional expressiveness, nor any interactive evaluations of the control interface.

(c) The paper does not explain or explote into how the time-varying emotion vector actually influences the TTS output beyond reporting improved scores. I understant that the model injects an emotion-conditioned branch at certain layers, but it’s a black box in terms of what it changes in the speech. For example, does a higher arousal primarily increase the pitch? or Does a sadness emotion elongate the speech or lower the volume? Are certain phonemes or words being pronounced differently under emotional stress?  One part of this is analyzing acoustic features: take identical text, synthesize it in different emotions or intensities, and then examine features like F0 (pitch contour), energy, duration, and spectral shape.

(d) The selective block strategy is derived empirically to inject the emotion only at certain transformer blocks and skip others (0,1,6,16). I appreciate this that is based on clever ablation ((skipping each layer to see its effect on WER and speaker similarity) but the paper doesn’t provide insight into why those layers are the ones to exclude beyond the numerical effect. In other words, what do those critical layers do, and why would adding emotion there be detrimental? There is a theoretical gap in understanding the model’s architecture.

(e)  T-VecTTS uses a continuous emotion representation (via Emotion2Vec and arousal–valence values), which should, in theory, allow a broad range of emotional expressions and even mixtures. However, the paper doesn’t show any visualization of this emotion space or examples of intermediate emotions. We don’t know if the model’s “emotion space” is well-behaved (e.g., is neutral in the center and various emotions spread out appropriately?) or if it only learned the discrete emotions present in training.

**Questions:**

(1) I think It can generalizes well within the training distribution, but extreme voices or languages outside the base model’s training set can lead to degraded quality or accent artifacts.

(2) I wonder if there are any instances  If the SER misidentifies or smooths over an emotional nuance, the TTS will faithfully reproduce that flawed signal.

(3)  Whether the model can interpolate or smoothly transition between emotions beyond the two-scenario concatenation (EMO-Change) tested.?

---

> ### Author Response · Authors · 2025-11-25
>
> ## W1.Generalization
> Thank you for raising this point. We clarify the behavior regarding backbone dependency.
>
> ### (a,b, Q1) Backbone-Dependent Behavior
> Our method is designed as an add-on module, and therefore, it inevitably inherits both the strengths and the limitations of the underlying backbone model. This also implies that any future improvements in the backbone architecture would directly benefit our method without requiring additional modifications.
>
> We will explicitly acknowledge this dependency and its implications in the revised paper for completeness and clarity.
>
> ### \(c\) Selective layer
> Our target is the ablating emotional controllability of *the flow matching-based TTS model*, which has a strong advantage in zero-shot voice cloning over the other mechanisms. The flow matching-based TTS not only includes [F5TTS] but also [Voicebox, EllaTE].
>
> ## W2,Q2. Dependence on SER
>
> Thank you for the comment. We would like to clarify that this limitation stems from the current SER model, not from T-VecTTS itself. Our method is SER-agnostic and can work with any recognizer. If a stronger SER that captures richer or discrete emotions becomes available, T-VecTTS can immediately benefit without changing the architecture.
>
> Additionally, during training, we use the SER’s emotion predictions as the conditioning signal and learn to reconstruct the target audio under that continuous emotion trajectory. At inference, we use the same SER model; therefore, the SER functions as a **mapping into an emotion space**, not as a strict classifier whose absolute recognition accuracy is crucial. Therefore, the method primarily depends on the consistency of the trajectory, not on perfectly correct labels.
>
> Even with a simple 2D arousal–valence signal, our results show that the model can already produce smooth and diverse emotional changes. Future improvements in SER will therefore directly translate into improved controllability for T-VecTTS rather than being a structural limitation.
>
>
> ## W3. Experiments
> ### (a) Audio samples
> Thank you for your concern. For the baseline model [F5-TTS], we used newly generated audio, but most other baselines were not publicly available, so we relied on the results reported in their original papers. To ensure fair comparison, we followed the evaluation protocols as faithfully as possible and used multiple random seeds to reduce variance.
>
> Specifically, we use `three different random seeds` for evaluation. Below, we clarify the evaluation protocol in detail.
>
> **For JVNV**, [EmoCtrl-TTS] reports using 1,615 samples, and we follow exactly the same protocol and the same sample set.
>
> **For EMO-Change**, the original code and sampling seeds were not released. Therefore, we reconstructed the evaluation setting by randomly concatenating utterances with different emotions.
>
> To improve statistical reliability and ensure fair comparison, we further increased the number of EMO-Change samples from 84 to 200, which yields more stable metrics under the same evaluation protocol.
>
> The final reported numbers are averages over three random seeds.
>
> We will include these details in the revised manuscript for clarity and reproducibility.
>
> [EmoCtrl-TTS] Laugh Now Cry Later: Controlling Time-Varying Emotional States of Flow-Matching-Based Zero-Shot Text-to-Speech (SLT2024)
>
>
> ### (b) Qualitative and Interactive Evaluation
> Thank you for the comment. We would like to clarify that our paper already provides qualitative human evaluations through **NMOS, SMOS, and EMOS**, which directly assess naturalness, speaker similarity, and emotional expressiveness. These MOS results offer subjective feedback on how the generated speech is perceived by listeners.
>
> Regarding interactive evaluations of the control interface, such tests are generally conducted for user-facing applications, rather than for model-level research contributions. Our work focuses on the model and its controllability, and the interface itself is not the main contribution.
>
> That said, we agree that interactive user studies could be valuable future work when deploying the method as part of a user-facing application, and we will mention this in the revised manuscript.

---

> > ### Author Response · Authors · 2025-11-25
> >
> > ## W\(c\). How the time-varying emotion vector influences acoustic features
> >
> > Thank you for the thoughtful question. We clarify the intended scope of our contribution and how the emotion trajectory affects the generated speech.
> >
> > Our goal is not to introduce a new form of manual, axis-specific emotion control, nor to claim interpretability of individual acoustic dimensions. Instead, the contribution of T-VecTTS lies in accurately reproducing a time-varying emotional trajectory extracted from reference speech, and demonstrating that this trajectory is reflected in the synthesized output.
> >
> > This effect is already observable in the EMO-Change experiments. Each sample consists of two contrasting emotions, and our model consistently produces a clear temporal transition (e.g., Angry → Calm). As shown in the supplementary qualitative results, these transitions naturally manifest as changes in emotions aligned with the reference trajectory, without requiring explicit feature-level supervision.
> >
> > While detailed acoustic analyses could provide additional insight, such an analysis is complementary rather than central to our claim. We will clarify this scope in the revised paper and discuss acoustic analysis as an interesting direction for future work.
> >
> > ## W(d) Insight on the selective block strategy
> >
> > Thank you for highlighting this point. Although transformer layers are not fully interpretable, prior studies show clear evidence of **layer-wise specialization** across domains. Motivated by this, we hypothesize that each layer contributes differently to emotion conditioning.
> >
> > To avoid relying solely on intuition, we adopt a **measurement-driven approach**: we empirically evaluate each layer’s sensitivity by injecting emotion at that layer and measuring its effect on WER and speaker similarity. This allows us to identify layers where emotion injection is beneficial and avoid layers where it is detrimental.
> >
> > We will incorporate this explanation into the manuscript to clarify the motivation and empirical grounding behind our design.
> >
> > ## Q3 Smooth transition between emotions
> > Yes, the model is capable of interpolating and producing smooth emotional transitions, as long as the reference audio provides a continuous emotional trajectory or interpolation.
> >
> > In the EMO-Change experiments, we used two clearly separated emotions primarily to make the perceptual differences more salient for human evaluators. For evaluating smoother transitions, we relied on the JVNV dataset, where emotions vary gradually over time. We reported frame-wise EMO-sim and arousal–valence similarity, both of which show consistent improvements over baselines.
> >
> > These results demonstrate that the model not only handles discrete emotion changes but also effectively follows continuous, fine-grained emotional trajectories.

---

### Official Review · Reviewer_m3wJ · 2025-11-01

**Soundness:** 3
**Presentation:** 2
**Contribution:** 1
**Rating:** 2
**Confidence:** 5

**Summary:**

This paper proposes fine-tuning F5-TTS with ControlNet for time-varying emotion control in text-to-speech. The framework can also be used for emotion-preserving speech-to-speech translation. Two key techniques are highlighted: (1) identifying layers that are important for preserving TTS quality and freezing them during training, and (2) applying control only on a subset of diffusion steps.

**Strengths:**

The paper included many details in Appendix. Reproducibility is good since relying on open source models.

The reported results look good, achieving state-of-the-art performance in compared baselines on two benchmarks, under the constraint of using much fewer data compared to the baselines.

**Weaknesses:**

The proposed techniques are of limited significance. The proposed techniques are specific to the emotion control task and the F5-TTS model, so it's transferrability and significance to other TTS paradigms (such as MaskGCT [3]) are not guaranteed.

The paper focus on the small data regime (400 hours), so it should focus on this scenario and compare with other fine-tuning methods with few data. There might be simpler and more lightweight solutions, such as training a guidance model and conduct classifier guidance [1], and fintuning the model with conditional layernorm [2].

In introduction (line 77) "without modifying its original parameters" seems a bit misleading. Since training with ControlNet seems nearly double the number of parameters. Again the solution seems a bit heavy on computation cost.

Since F5-TTS does not support Japanese, the evaluations on JVNV S2ST relies on converting Japanese to Romaji. So the training inference mismatch is severe on JVNV S2ST test set. How can the NMOS and SMOS be improved with T-VecTTS fine-tuning?

For objective evaluation, please report more details as for how the results are calculate. Did you conduct repeated evaluation with different random seeds?

[1] EmoDiff: Intensity Controllable Emotional Text-to-Speech with Soft-Label Guidance
[2] AdaSpeech 4: Adaptive Text to Speech in Zero-Shot Scenarios
[3] MaskGCT: Zero-Shot Text-to-Speech with Masked Generative Codec Transformer

**Questions:**

Please see section on weaknesses.

---

> ### Author Response · Authors · 2025-11-24
>
> We carefully correct the concerns as follows:
>
> ## W1. Transferrability to other TTS paradigms
> Our work focuses on evaluating emotional controllability within *flow-matching–based TTS models*, which are known for their strong zero-shot voice cloning capability. This family of models includes not only F5-TTS but also Voicebox and EllaTE.
>
> We note that architectures outside the flow-matching paradigm—such as [MaskGCT]—are beyond the scope of this work.
>
> [MaskGCT]: Zero-Shot Text-to-Speech with Masked Generative Codec Transformer
>
> ## W2. Comparison with a small, lightweight solution
> Both of [EmoDiff] and [AdaSpeech 4] support only *utterance-level* control and do not provide *time varying* emotional control.
>
> [EmoDiff]: Intensity Controllable Emotional Text-to-Speech with Soft-Label Guidance [AdaSpeech 4]: Adaptive Text to Speech in Zero-Shot Scenarios
>
> ## W3. Possibility of misleading terms
> Thank you for the constructive feedback. We agree that the phrase “without modifying its original parameters” may be misleading. In practice, we freeze the original model and train only the conditioning branch, which is a trainable copy derived from the original architecture. This design choice allows the conditioning branch to inherit the TTS processing capability of the base model.
>
> We revised the wording in the paper to make this clearer.
>
> ## W4. F5TTS with Japanese
> Thank you for raising this concern.
> Although our backbone model, F5-TTS, is trained only on English, it supports cross-lingual zero-shot voice cloning and can use speech from other languages as reference audio. However, it struggles with time-varying emotional control and often produces utterances with a monotonic emotional trajectory, which leads to unnatural or robotic-sounding speech.
>
> T-VecTTS preserves the cross-lingual zero-shot capability by freezing the original network, while additionally providing time-varying emotional control. We believe that introducing natural emotion transferability helps resolve the monotonicity issue.
>
> For clarity, we incorporated this explanation into the paper.
>
> ## W5. More details for the evaluation
>
> We use **three different random seeds** for evaluation. Below, we clarify the evaluation protocol in detail.
>
> For **JVNV**, EmoCtrl-TTS reports using 1,615 samples, and we follow exactly the same protocol and the same sample set.
>
> For **EMO-Change**, the original code and sampling seeds were not released. Therefore, we reconstructed the evaluation setting by randomly concatenating utterances with different emotions.
>
> To improve statistical reliability and ensure fair comparison, we further increased the number of EMO-Change samples from 84 to 200, which yields more stable metrics under the same evaluation protocol.
>
> The final reported numbers are averages over three random seeds.
> We will include these details in the revised manuscript for clarity and reproducibility.
>
> - [EmoCtrl-TTS] Laugh Now Cry Later: Controlling Time-Varying Emotional States of Flow-Matching-Based Zero-Shot Text-to-Speech (SLT2024)

---

### Official Review · Reviewer_MyPE · 2025-11-03

**Soundness:** 3
**Presentation:** 3
**Contribution:** 2
**Rating:** 4
**Confidence:** 4

**Summary:**

This paper adds time-varying emotion control to a flow-matching TTS base model (e.g., F5-TTS) without full fine-tuning. The authors freeze the base and attach a small, trainable conditioning branch connected via zero-convolutions; the branch ingests a frame-level arousal/valence sequence from a SER model and injects it into selected DiT blocks. They also identify an emotion-specific flow-step interval and apply emotion control only in that window to reduce WER while improving emotional expressiveness.

**Strengths:**

- Freezes the backbone and adds a compact trainable branch tied by zero-conv, making it easy to work on strong flow-matching TTS models.
- The paper follows the EmoCtrl-TTS protocol (EMO-Change, JVNV S2ST) and reports the same metrics (WER, AutoPCP, Emo-SIM, Aro-Val SIM).
- Clear demonstration of intra-utterance transitions (e.g., sad→happy) using the reference emotion trajectory, similar with EmoCtrl-TTS
- Uses ~400 h public emotional speech and shows that emotion-rich data matters more than raw hours (LibriSpeech control).

**Weaknesses:**

- Limited conceptual novelty over EmoCtrl-TTS. The main novelty is how emotion is injected (a zero-conv conditioning branch + flow-step scheduling + block selection), which feels incremental relative to prior time-varying emotion control via full fine-tuning.
- Gains are strongest on emotion similarity; SIM-o/AutoPCP/WER are more mixed (and authors attribute some WER gaps to cross-lingual S2ST and base-model language coverage).

**Questions:**

- Why do SIM-o / WER / AutoPCP sometimes degrade? Any insights on it.
- In Figure 2, is it correct to include the ‘Noisy speech’ (training) and ‘Noise’ (inference) blocks?
- Training data: which six datasets? You mentioned "To overcome this limitation, we constructed a training dataset by combining six datasets", could you explain more regarding the details.
- A follow-up question, any studies over the training data, which one is more important to make the performance better?

---

> ### Author Response · Authors · 2025-11-24
>
> We carefully address the questions and concerns as follows:
>
> ## W1: Novelty over EmoCtrl-TTS
> We appreciate the reviewer’s comment. Obtaining high-quality emotional speech data is difficult, and many existing methods depend on proprietary datasets or full fine-tuning.
>
> Our method instead combines zero-conv conditioning, flow-step scheduling, and selective block control to achieve time-varying emotion control using only public data and without any additional fine-tuning. Although each component is simple, their integration provides a practical and data-efficient alternative to training-heavy approaches.
>
> Despite this lightweight design, our method still delivers competitive performance, indicating that it offers a meaningful complementary contribution in scenarios where curated emotional datasets are unavailable.
>
> ## W2, Q1: SIM-o, WER AutoPCP sometimes degrades?
> We are not entirely sure which specific configuration the reviewer is referring to regarding the reported degradation, but we offer the following reasonable interpretation:
> - Degradation relative to other competitors:
> We conjecture that this comes from the characteristics of the baseline model. Our method is training-free and preserves the pretrained model’s behavior while adding time-varying emotional control. In other words, our results largely inherit the strengths and weaknesses of the base model (F5-TTS), which can lead to slight degradation on certain metrics and improvements on others.
> - Degradation within the baseline (F5-TTS):
> We observed minor degradation in AutoPCP under the JVNV cross-lingual setting. We believe this is due to the inherent difficulty of cross-lingual prosody transfer, where pitch and rhythm patterns differ significantly between source and target languages. This makes prosody-alignment metrics such as AutoPCP particularly sensitive.
>
> ## Q2: Noisy speech & noise as inputs
>
> We clarify how noisy inputs are used during training and inference.
> During training, we create noisy speech by adding Gaussian noise to the target speech, and this noisy signal is then fed into the network.
>
> During inference, however, we generate emotional speech by iteratively denoising from random Gaussian noise, which corresponds to `“Noise x_0”` shown in Fig. 2(b).
>
> ## Q3: Details of training dataset
>
> We kindly address your confusion with the following details of the training dataset.
>
> | Dataset     | Duration (hours) | # of Speakers | # of Emotions | Modality     |
> | - | -| -| - | - |
> | IEMOCAP     | 12.44| 10 | 8  | Audio-Visual |
> | ESD         | 29.07| 10  | 5 | Audio        |
> | MSP-Podcast | 319.15| ~500           | 7             | Audio        |
> | Expresso    | 11.26| 8             | 4             | Audio        |
> | RAVDESS     | 1.48| 24            | 8             | Audio-Visual |
> | EARS        | 54.96            | 20            | 6             | Audio        |
> | **Total**   | **428.36**       | —             | —             | —            |
>
> We provide the six datasets in Section 4.2 with citations, and further include more details of each training dataset in the paper.
>
>
>
> ## Q4: What is a good training dataset?
>
> We thank the reviewer for the valuable suggestion regarding dataset analysis. To better understand the relationship between dataset size/type and performance, we conducted additional ablation studies under the same compute budget:
>
> - Emotion speech data (400h → 200h)
> - Non-emotional data ([LibriSpeech] 400h)
>
> As shown in the table below, reducing the emotion dataset from 400h to 200h leads to a slight increase in WER and slightly lower emotion similarity. However, the performance drop remains modest.
>
> In contrast, when using non-emotional LibriSpeech data—a read-speech dataset with minimal emotional variation—we observed a degradation in emotional metrics, particularly Emo-SIM and Aro-Val SIM. This highlights the importance of emotion-rich supervision for our task.
>
> All variants were trained for the same number of total steps to ensure a fair comparison under equal compute cost.
>
> [LibriSpeech]: An ASR corpus based on public domain audio books
>
>
> ### JVNV S2ST
> | Methods|Training data|Backbone|SIM-o  $\uparrow$|WER $\downarrow$|AutoPCP $\uparrow$|Emo-SIM	$\uparrow$|Aro-Val SIM	$\uparrow$|
> |-|-|-|-|-|-|-|-|
> |F5-TTS | Emilia (95k)|N/A|0.459|**4.5**|**2.87**|0.684|0.627|
> | EmoCtrl-TTS (reproduce)|Public emotion speech (400h)|F5-TTS|0.443|7.8|2.31|0.692|0.659|
> |TTS-CtrlNet (Ours)|Public emotion speech (400h)|F5-TTS|**0.464**|5.4|2.36|**0.751**|**0.742**|
>
> ### EMO-Change
> |Methods|Training data|Backbone|SIM-o  $\uparrow$|WER $\downarrow$|AutoPCP $\uparrow$|Emo-SIM	$\uparrow$|	Aro-Val SIM	$\uparrow$|
> |-|-|-|-|-|-|-|-|
> |F5-TTS| Emilia (95k)|N/A|0.579|2.9|3.62|0.692|0.845|
> |EmoCtrl-TTS (reproduce)| Public emotion speech (400h)|F5-TTS|0.572|4.4|3.69|0.709|0.851|
> |TTS-CtrlNet (Ours)| Public emotion speech (400h)|F5-TTS|**0.589**|**0.6**|**3.68**|**0.724**|**0.864**|

---

### Author Response · Authors · 2025-12-04

As the PDF update period is ending soon, we kindly invite reviewers who have not yet joined the discussion to share any remaining questions or concerns.

We appreciate all the reviewers for acknowledging our strengths:

- Practical, efficient adaptation: adds fine-grained, time-varying emotion control while preserving the base model’s zero-shot capability and speech quality (naturalness/intelligibility). [MyPE, VQtb, fYW3]
- Clear qualitative evidence: convincing intra-utterance emotion transitions (e.g., sad→happy) guided by reference emotion trajectories. [MyPE, VQtb]
- Fair evaluation protocol [MyPE]
- Highlights the importance of emotion-rich datasets  [MyPE]
- Strong results with less data: state-of-the-art performance on two benchmarks, under the constraint of using substantially less training dataset [m3wJ, VQtb]
- No full fine-tuning enables avoiding the heavy cost of full model fine-tuning [VQtb]
- The proposed emotion-specific flow-step is well-motivated and has broader implications for controlling other attributes [fYW3]
- Reproducibility and detailed analysis [m3wJ]
- Multiple ablations (flow step range, selective blocks, control scale, emotion window size) that evaluate design choices and tradeoffs (emotion vs. WER, inference time). [fYW3]

And providing ingredients to strengthen our paper:
- Additional comparisons and positioning. [MyPE,m3wJ]
- Clarification on metric behavior. [MyPE, fYW3]
- Clarification on how noisy inputs are used during training and inference. [MyPE]
- More details on training data and evaluation [MyPE], including experiment details (audio samples, qualitative/interactive evaluation). [VQtb]
- Out-of-distribution language as a reference audio. [m3wJ]
- Generalization analysis (backbone-dependent behavior, selective layers/blocks, and dependence on SER). [VQtb]
- Details of the Experiments (Audio samples, Qualitative and Interactive Evaluation). [VQtb]
- How the time-varying emotion vector influences acoustic features. [VQtb]
- Insight on the selective block strategy. [VQtb]
- Smooth transition between emotions. [VQtb]
- Statistical/presentation fixes: confidence intervals and presentation issues. [fYW3]


In the rebuttal, we have carefully addressed the comments so that the reviewers can anticipate our high-quality camera-ready version. If there are any other questions or concerns, please feel free to post another comment. We are grateful to all reviewers for their constructive feedback, and we also thank Reviewer fYW3 for the helpful follow-up discussion and the updated scores. [fYW3]

---

### Meta-Review · Area_Chair_hrst · 2026-01-07

**Summary:**

Overall, the paper is being judged as a “nice engineering package” for time-varying emotion control on top of a flow-matching TTS backbone, but there’s real disagreement on whether it’s actually a meaningful step beyond EmoCtrl-TTS (incremental vs. practical contribution).
The biggest decision-driving concerns were (i) limited conceptual novelty / limited transferability beyond F5-TTS and flow-matching, and (ii) trust in the experimental evidence (small test set artifacts like WER=0, missing stats originally).
The rebuttal helped on the “do we trust the numbers?” front (explaining WER=0 and adding MOS confidence intervals), and it also clarified training data and evaluation protocol, but it didn’t fully eliminate the “this feels narrow / incremental” critique from the more negative reviewer.

**Reviewer Concerns:**

Addressed (or mostly addressed) by the rebuttal

MyPE: The authors gave a clear list of the six public emotion datasets (with hours / speakers / emotions) and added an extra ablation showing emotion-rich data matters more than just hours (vs. LibriSpeech).

MyPE (confusion about noise blocks in Fig. 2): They explained “noisy speech” during training vs. “Gaussian noise” as the diffusion starting point at inference, so the diagram choice is defensible.

m3wJ (misleading “no parameter modification” wording): The authors admitted the phrasing was misleading and clarified it’s “freeze backbone + train a conditioning branch/copy,” and they said they revised the wording.

m3wJ (evaluation reliability / seeds): They explicitly said they use 3 random seeds, follow JVNV’s protocol/sample set, and expanded EMO-Change to 200 samples for stability, which addresses “how are results computed?” concerns.

fYW3 (WER=0 + stats): They explained why WER=0 happens (tiny, clean, fixed-prompt RAVDESS setup; Table 5 copied from EmoCtrl-TTS), and they added MOS confidence intervals; the reviewer accepted this and raised their overall rating to 6.

Still outstanding / only partially addressed

Core novelty / significance: Both MyPE and m3wJ basically argue the main ideas (where/when to inject control; scheduling) are incremental and pretty tied to F5-TTS / flow-matching, and the rebuttal mostly reframes this as “practical + public data + no full finetune,” which may not convince a skeptical audience.

Comparisons vs. other “lighter” controls (m3wJ): The authors argue EmoDiff/AdaSpeech4 are utterance-level (not time-varying), but they still don’t really benchmark against a broader set of lightweight control baselines within the same paradigm, so the “maybe there’s an even simpler solution” concern isn’t fully killed.

Mechanistic insight / analysis depth (VQtb): The rebuttal essentially says “interpretability/acoustic-feature analysis is future work,” which is reasonable, but it means those analysis-oriented asks remain unaddressed.

Generalization risk from tiny test setup (fYW3): Even after agreeing WER=0 is explainable, the reviewer explicitly kept concerns about generalization due to the small test set.

**Reviewer Scores:**

Reviewer MyPE: They were already close to the line and mainly wanted clarity on data composition + why some metrics drop + what dataset matters; the rebuttal directly answered those and added extra ablations, so a small bump (to weak accept) seems plausible.
Reviewer m3wJ: This reviewer’s stance is fundamentally “too narrow / not transferable / not compared against the right lightweight baselines,” and while the rebuttal clarifies wording, scope (flow-matching family), and seeds, it probably doesn’t fix the core “significance” objection; at best it nudges to a 3.
Reviewer VQtb: The rebuttal gives reasonable answers (SER-agnostic framing, scope clarification, and evaluation protocol explanation), but many of their asks were “deeper analysis” rather than quick fixes, so this is more likely to stay borderline, maybe a slight bump if they value the clarifications.
Reviewer fYW3: They already participated and explicitly updated their overall rating to 6 after the authors clarified WER=0 and added statistical intervals, so the “full discussion” counterfactual is basically known here

---

### Decision · Program_Chairs · 2026-01-26

Reject